# Results and Strategies for a Diversity-Oriented Public Health Monitoring in Germany

**DOI:** 10.3390/ijerph19020798

**Published:** 2022-01-12

**Authors:** Carmen Koschollek, Katja Kajikhina, Susanne Bartig, Marie-Luise Zeisler, Patrick Schmich, Antje Gößwald, Alexander Rommel, Thomas Ziese, Claudia Hövener

**Affiliations:** Robert Koch Institute, 13353 Berlin, Germany; KajikhinaK@rki.de (K.K.); BartigS@rki.de (S.B.); ZeislerM@rki.de (M.-L.Z.); SchmichP@rki.de (P.S.); GoesswaldA@rki.de (A.G.); RommelA@rki.de (A.R.); ZieseT@rki.de (T.Z.); HoevenerC@rki.de (C.H.)

**Keywords:** public health monitoring, public health reporting, migration, diversity-oriented, discrimination, core indicators

## Abstract

Germany is a country of immigration; 27% of the population are people with a migration background (PMB). As other countries, Germany faces difficulties in adequately including hard-to-survey populations like PMB into national public health monitoring. The IMIRA project was initiated to develop strategies to adequately include PMB into public health monitoring and to represent diversity in public health reporting. Here, we aim to synthesize the lessons learned for diversity-oriented public health monitoring and reporting in Germany. We also aim to derive recommendations for further research on migration and health. We conducted two feasibility studies (interview and examination surveys) to improve the inclusion of PMB. Study materials were developed in focus groups with PMB. A systematic review investigated the usability of the concept of acculturation. A scoping review was conducted on discrimination as a health determinant. Furthermore, core indicators were defined for public health reporting on PMB. The translated questionnaires were well accepted among the different migrant groups. Home visits increased the participation of hard-to-survey populations. In examination surveys, multilingual explanation videos and video-interpretation services were effective. Instead of using the concept of acculturation, we derived several dimensions to capture the effects of migration status on health, which were more differentiated. We also developed an instrument to measure subjectively perceived discrimination. For future public health reporting, a set of 25 core indicators was defined to report on the health of PMB. A diversity-oriented public health monitoring should include the following: (1) multilingual, diversity-sensitive materials, and tools; (2) different modes of administration; (3) diversity-sensitive concepts; (4) increase the participation of PMB; and (5) continuous public health reporting, including constant reflection and development of concepts and methods.

## 1. Introduction

### 1.1. Migration in Germany

According to the International Organization for Migration (IOM), the number of international migrants has been rising during the past 50 years, with most migrants residing in Asia and Europe in 2019 [1]. Germany, as the second top destination country after the United States of America [1], is a country of immigration. In 2020, more than every fourth resident (26.7%) had a so-called migration background, meaning that the person or at least one of their parents were born with a nationality other than German [2]. A total of 62.1% of this group were born abroad and migrated themselves, while 52.8% have German nationality [2]. People with a migration background form a very heterogeneous subpopulation, differing (for example) in terms of time spent living in Germany, reasons for migration, living situation before and after migration, legal status, and many other factors, which are linked to chances and also barriers in accessing social resources, such as education, labour market, or health-care services [3,4]. In regards to health-care utilization, and health outcomes differences are observed [5,6,7,8,9]. Hence, it is difficult to use the statistical category *migration background* as a single predictor variable when analysing differences in health outcomes. To facilitate analyses of differences in health outcomes that do not solely consider the statistical category *migration background*, it is essential to include people with migration history into public health monitoring, namely (a) equivalent to their proportion of the population and also (b) considering the heterogeneity of this group.

### 1.2. Inclusion of People with Migration History into Public Health Monitoring

As other countries [10,11,12,13,14], Germany faces difficulties in adequately including hard-to-survey populations, amongst others people with migration history, people with lower educational levels, and people living in deprived areas, into nationwide public health monitoring. Several studies identifying ways on how to reach hard-to-survey populations either focus on specific sub-populations [15,16,17,18] and areas [16,19,20] or on inclusion into specific study types, for example, clinical trials [21] or intervention studies [15,22]. Recommendations on how to include hard-to-survey populations into a comprehensive public health monitoring are scarce [23,24,25].

Register-based random samples are drawn for health examination surveys conducted by the Robert Koch Institute (RKI), which is the Federal Public Health Institute in Germany, including oversampling of people without German nationality to account for the lower response rates among people with migration history. Other measures to adequately include this group into the German Health Interview and Examination Survey for Children and Adolescents (KiGGS, baseline study 2003–2006; wave 1 2009–2012 (only interview survey); wave 2 2014–2017) as well as the German Health Interview and Examination Survey for Adults (DEGS, 2008–2011) included the translation of study materials and questionnaires, home visits, and partial contact with migrant-specific media.

In the KiGGS baseline survey, study materials and questionnaires were translated into six languages (i.e., Arabic, English, Russian, Serbo-Croatian, Turkish, and Vietnamese) [26]. An onomastic procedure was applied to assign the names of study persons to these languages [27,28]. Participants with insufficient German-language skills were allowed to bring a lay interpreter to the examination appointment. If no lay interpreter was available, then important information on the examinations and a short paper-based questionnaire to exclude contraindications in the six languages and additionally in Italian were offered [26]. Home visits took place for non-responders [29]. In the KiGGS baseline survey, 8.4% of participants (unweighted) did not have German nationality compared to 8.8% of children and adolescents in official statistics (2004, [30]). The response rate for German nationals was 68% compared to 51% for non-German nationals [31].

For KiGGS wave 2, study materials and also questionnaires were provided in four languages (i.e., English, Russian, Serbo-Croatian, and Turkish) [32], and the onomastic procedure was applied. In addition, the field staff received intercultural training, and non-responding study persons were visited at home prior to the field phase for medical examinations. Here, a particular focus was put on families with a migration history (non-German nationality) to assure their inclusion. Home visits doubled the likelihood of survey participation in migrant families and thus proved to be the most effective strategy to increase the response rate [32]. Due to an increase of immigration in 2015, especially from Syria and Iraq (among others), a short questionnaire was developed and translated into Arabic and English to grasp basic data on the health status of recently migrated children and adolescents [32]. In KiGGS wave 2, 3.7% of participants (unweighted) did not have German nationality compared to 7.0% of children and adolescents in official statistics (2013, [33]). The response rates were also lower in non-German nationals (27.3% vs. 41.5%) [34]. Overall, the integration of children and adolescents and their families into the KiGGS survey worked well.

In DEGS (German Health Interview and Examination Survey for Adults, 2008–2011), the questionnaires were translated into four languages (i.e., English, Russian, Serbo-Croatian, and Turkish) [35]. However, sufficient German-language skills were required to provide informed consent towards mainly German-speaking study personnel conducting the examinations, which resulted in a systematic exclusion of people with a German proficiency level under B1 (CEFR; Common European Framework of Reference for Languages). Although other measures included home visits to non-responders, this measure did not focus on people with a migration background specifically [35]. In DEGS, the participation of migrants was lower. For example, 8.2% of participants (unweighted) had an own history of migration compared to 13.7% in official statistics (2009, [36]), 4.6% (unweighted) had a nationality different than German compared to 7.8% in the adult population (2009, [36]), and particular migrant groups were underrepresented (e.g., people with Turkish nationality) [37].

Health interview surveys conducted by the RKI, such as the regularly conducted survey GEDA (German Health Update), are mainly administered by telephone (random digit dialling) and in German. Therefore, people with non-German dialling numbers and also people without sufficient German-language skills are systematically excluded. In GEDA 2014, a proportion of 7.2% of adult participants had an own history of migration compared to 15.2% according to official statistics (2014, [38]), and 3.2% were of non-German nationality in contrast to 9.3% within the adult population (2014, [38]). We observe similar trends in GEDA 2019/2020: an own history of migration was reported by 9.0% of participants compared to 18.4% of the population (2019, [39]); 3.9% of participants were of non-German nationality compared to 12.5% of the adult population (2019, [39]). Health interview surveys often underrepresent other groups (e.g., people with lower socioeconomic position), which also might be associated with the underrepresentation of some migrant groups because the risk of poverty is increased for people with migration background [40].

Even though all these different measures have had positive effects, there still seem to be further barriers for people with migration background who wish to participate in health surveys. Expert interviews revealed that besides sociodemographic, cultural, and language barriers, challenges, fears, as well as structural and practical barriers may play a role [41]. They also identified some strategies to improve the inclusion of people with a migration background into health surveys that address multiple dimensions, such as communication, trust, and participation. Furthermore, it is necessary to consider these aspects during all stages of the research. This also means that participation of people with a migration background in setting up study designs, as well as a diversity-sensitive research culture, are crucial and important to build trust [41].

### 1.3. Public Health Reporting

As mentioned earlier, the category *migration background* is not sufficient to report on differences in health status and access to health-care in migrant groups because it is not able to adequately address the diversity in migrant populations [42,43]. Methodological and theoretical concepts and survey instruments therefore need to be critically revised and further developed to analyse health inequalities that are associated with differing health outcomes. Additionally, core indicators on migrant health are needed to enable a systematic and regular public health reporting [44,45,46], as well as international comparability of health status and associated factors in migrant populations.

### 1.4. The IMIRA Project: Improving Health Monitoring in Migrant Populations

To address these challenges, the *Improving Health Monitoring in Migrant Populations* (IMIRA) project was initiated at the RKI in 2016. The aims of this project were (a) to develop strategies to improve the inclusion of people with migration background into public health monitoring, (b) to revise concepts and survey instruments, and (c) to further develop public health reporting on people with a migration background. 

The aim of this paper is to synthesize the findings of IMIRA and the lessons learned for public health reporting in Germany in order to derive recommendations for further research on migration and health. Given that this is a synthesis paper, some of the findings have been published elsewhere and will be summarized here, while others are original results.

## 2. Materials and Methods

### 2.1. Feasibility Study: “Interview Survey”

To test different recruitment strategies and to increase response rates, in particular for hard-to-survey groups, we conducted a sequential mixed-mode health interview survey among people with Croatian, Polish, Romanian, Syrian, or Turkish nationality in the federal states of Berlin and Brandenburg, Germany, from January to May 2018. Register-based random samples were drawn by nationality out of seven selected primary sampling units. Study materials and questionnaires were translated and made available in the Arabic, Croatian, German, Polish, Romanian, and Turkish languages. Study materials were developed in focus groups with representatives of the respective migrant groups. In the first step of the sequential design, study persons (*n* = 9068) received a bilingual invitation letter and were invited to answer a multilingual online questionnaire. In a first reminder letter sent to those study persons who did not participate or refused survey participation, telephone interviews with native speakers were offered to the study persons by calling a hotline. A second reminder letter then announced a home visit to a subsample of study persons (*n* = 1822) with Romanian, Syrian, or Turkish nationalities in Berlin. During the home visits, in two different subsamples, either the telephone number of a study person willing to participate was obtained to conduct a bilingual telephone interview, or a face-to-face-interview was performed directly at the participants’ home [47].

After completing the data collection, focus group discussions took place with the interviewers who conducted the home visits in the last phase of the study to discuss the barriers and difficulties that they had faced in the field and to collect any of the strategies that they had used to convince study persons to participate.

### 2.2. Feasibility Study: “Examination Survey”

To overcome language gaps in health examination surveys and to assure informed consent independent of German-language skills, a feasibility study (“examination survey”) was conducted. The participants were recruited via convenience sampling and included persons with Polish, Syrian, or Turkish migration backgrounds. Recruitment was based on the following criteria: the participants were (a) living in Berlin; (b) speaking either Arabic, Polish, or Turkish; and (c) speaking German on a level B1 or below (CEFR; Common European Framework of Reference for Languages). Sampling was further based on gender and age groups. The participants were invited to take part in an examination in a temporary study centre that was established at RKI. All of the study materials that were handed out to participants (i.e., information flyer, consent form, data privacy statement, and results report) were bilingual. 

In order to inform participants about the study and to explain examination procedures, we showed standardized videos in the relevant languages. Study personnel utilized a video interpreter service at two steps: first, to address questions, review a short questionnaire on possible contraindications, and to obtain informed consent; and second, to present the results of the examinations and to conduct a standardized short interview to evaluate the participants’ study participation overall. Focus groups were conducted with selected study participants to get more in-depth information on the barriers and motivation to participate, as well as on suggestions to improve the acceptance of health examination surveys in migrant populations.

### 2.3. Critical Revision of Concepts and Survey Instruments

In order to identify concepts used in epidemiological research on migration and health that go beyond the statistical category *migration background,* we screened recommendations on capturing migrant health as well as related questionnaires [48,49,50,51]. The concepts that were identified as relevant included (among others) acculturation and discrimination. Furthermore, the concept *migration background* was critically revised regarding its applicability in future studies based on the existing literature and expert debate.

A systematic review was conducted following the PRISMA statement [52,53] concerning the concept of acculturation because it has been widely used in research on migration and health. Focus was put on its use in epidemiological research, its theoretical embedment concerning the relationship between migration and health, its operationalization, and on recommendations for data collection on acculturation and health [52].

A scoping review was conducted because interpersonal and structural discrimination can have an impact on health, while people facing discrimination are not necessarily represented by the statistical category *migration background*. Quantitative studies were included that used racial discrimination as an anchor category [53,54,55]. Approaches to operationalize and analyse discrimination as a determinant of health in health surveys were compared. In addition, recommendations from counselling centres and other civil society institutions focusing on discrimination were considered [53,55]. Based on the results of the review, and considering methodological and ethical standards, an instrument on subjectively perceived discrimination was developed to be used within the health surveys at RKI [53]. 

All of the improved or newly integrated concepts and instruments regarding determinants of migration status and discrimination were cognitively pre-tested [56,57,58] to assure a common understanding in German and in five further languages (i.e., Arabic, Croatian, Italian, Polish, and Turkish). Instruments were further adapted at points where the cognitive pretesting revealed serious discrepancies in understanding [53]. 

### 2.4. Public Health Reporting Concept and Development of Core Indicators

To develop a concept to integrate data on migrant health into regular public health reporting, a mixed-methods approach was applied. On an international level (i.e., EU, including candidate countries, and OECD countries), national public health institutes were contacted and invited to take part in an online-survey from mid-March to mid-April 2018. In addition, a systematic Internet search was conducted for those countries not responding to the survey. This process aimed to reveal some of the strategies of public health reporting to identify best-practice examples of a migration-sensitive health reporting.

To determine core indicators for regular public health reporting on people with a migration background, the relevant fields of action were first identified based on WHO EURO guidelines for the health of migrants [48,59,60], and relevant topics were then assigned. The topics were then reviewed internally at RKI and were afterwards reviewed by the IMIRA Advisory Board members. Indicators were developed for the topics. Core indicators were derived by the availability of representative data and a comparison of established indicator systems to ensure national and international connectivity [61]. 

## 3. Results

### 3.1. Feasibility Study: “Interview Survey”

Before starting the survey, a focus group discussion was held to investigate some of the motivations for study participation and to further develop study materials. Three female and two male participants aged between 22 and 45 years, who were born in Croatia, Poland, Romania, Syria, and Turkey took part. The participants preferred to be addressed in writing, preferably via e-mail, instead of being approached via telephone or in the streets. The motivation for taking part in a survey was first of all influenced by the survey topic and by the way in which people feel that their opinion is heard. In addition, seriousness and the expenditure of time play a significant role. Incentives were assessed as an important tool to raise motivation for participation, while money in cash was preferred compared to vouchers. Written study invitations should not include too much text, while what will happen during and after participation should be clearly stated. Contact information and possible incentives should also be highlighted. With regards to the RKI, the wording “The Robert Koch Institute is a Federal Institute within the portfolio of the German Federal Ministry of Health” raised discomfort, and the IMIRA logo, including the phrase “*Improving Health Monitoring in Migrant Populations*”, was seen to be critical, because not everyone who is approached might identify as a *migrant*. Taking this feedback into account, the study materials were revised, and the study persons were invited to the survey.

Overall, 1190 people took part in the survey; the overall response rate was 15.9%. The response rate was highest in the group with Syrian nationality (24.3%), while it was lowest among those with Turkish nationality (8.6%) [62]. We recorded a proportion of 32.7% of quality neutral losses within the group with Romanian nationality (i.e., a third of the register-based sample was out of the sampling frame of the survey). Throughout all nationalities, most participants took part in online mode (79.9% overall) (Table 1). Participants with Croatian nationality preferably took part in German language (76.4%), while Syrians most often answered in Arabic (80.9%). Home visits, which were only offered in Berlin and in the groups with Romanian, Syrian, and Turkish nationality, increased participation rates +5.4% in the group with Turkish nationality and +7.3% in the group with Syrian nationality. During home visits, more participants with low levels of education and participants reporting a moderate to very bad self-rated health could be included in the sample than was possible in online and telephone mode (via hotline) only (Table 1). More detailed results are published in [62]. 

Focus groups that were held with the interviewers showed that participants appreciated interviewers with the same country of origin and interviews in their preferred language during home visits. The interviewers also stated that an effective argument to convince potential respondents to participate was to highlight that the results of the survey could eventually improve health services for that particular population group in Germany. The interviewers also highlighted that establishing trust was essential within this dialogue and underscored the importance of hiring diverse staff.

### 3.2. Feasibility Study: “Examination Survey”

Of 259 participants, 89 spoke Polish, and 85 spoke Arabic or Turkish each; 51% (*n* = 133) were female. The median age of the participants was 46.7 years. Of all participants, 99% confirmed feeling well prepared (16%) or very well (83%) for the examinations by watching the videos beforehand. The study personnel confirmed that the instructions in the videos were followed exactly or mostly in most cases, except for spirometry (Figure 1).

The video interpreter service was utilized 517 times, with an average call-time of 13 min. A total of 254 out of 257 participants stated that this service was very helpful or helpful, and all participants said that they appreciated the service. The quality of interpretation was rated as good. Meanwhile, Arabic-speaking participants indicated many facets of Arabic language, especially if interpreters themselves did not originate from Syria. In 26% of video interpreter service utilizations, the study personnel recorded difficulties in terms of Internet connection problems due to the technical set-up. The recorded difficulties were that connections broke down, the video pictures froze or were pixelated, and speech was slowed down.

Focus group discussions with participants showed that a health examination is seen as a check-up. Therefore, some participants would have wished for more “serious” examinations, such as taking blood samples or examinations on bone mass (especially for older participants). Another suggestion was to receive more in-depth examinations instead of a financial incentive. The participants appreciated that the study personnel took time for each participant, especially emphasizing the contrast to ambulatory health care, where time is often restricted. Scheduling conflicts were sometimes a barrier to participation. Consequently, it was suggested to propose holding appointments on weekends for people who work during the weekdays. Discussions also arose with regards to gender differences between participants, study personnel, and video interpreters. The participants identified some especially hard-to-survey groups and recommended solutions to include these groups, including home visits or pick-up services for the elderly. For parents with young children, they suggested to include the children in the examinations (as a check-up) or to offer child-care during the visit at the study centre. To better integrate young people, they proposed advertising at public places (e.g., on the metro) and to offer more specific incentives to young people (e.g., trial subscriptions at gyms).

### 3.3. Critical Revision of Concepts and Survey Instruments

The definition and operationalization of *migration background* as a statistical category to capture residents with a migration history in Germany has undergone a series of adjustments since its introduction to the official statistics in 2005. Objectifiable characteristics, such as nationality, nationality at the time of birth, or the existence of an own immigration history, serve as a basis for this concept [42,64]. In this way, a multitude of different life situations and perspectives are combined into one proxy variable. However, relevant health determinants, such as experiences of racial discrimination, cannot be mapped on the basis of this proxy variable [65]. 

Overall, 267 publications were included in the systematic review of acculturation, of which 49.4% reported on population-based surveys, and 50.6% were smaller studies using convenience samples [52]. Most studies were conducted in the United States (77.2%). The most common migrant groups under study were Asians (43.4%) or Hispanics (32.1%), while 3.0% of studies considered multiple migrant groups. The health outcomes that were investigated include mental health (27.7%), health behaviour (26.2%), or physical health (24.2%). Acculturation was measured by acculturation scales in 46.4% of studies and in 43.1% by proxies, while 10.5% of studies used both. These studies made use of 57 different acculturation scales, and 33 proxies were identified, which were assigned to the following four dimensions: language (*n* = 14), migration history (*n* = 11), ethnicity (*n* = 4) and social environment/culture (*n* = 4) [52]. Half of the studies (55.1%) included a definition of acculturation, 38.2% stated that acculturation is an important concept when analysing migration and health and 6.7% provided no definition at all [52]. In summary, the concept of acculturation shows not only pronounced inconsistencies regarding its theoretical fundamentals but also in its operationalization and measurement. Hence, instead of building an acculturation score across these highly diverse dimensions, differentiated concepts, such as feeling of belonging, social support, language skills, and migration history, seem more appropriate to be analysed as health-relevant factors for migrant populations [52,53]. Concrete items are recommended to describe these dimensions in surveys [53].

The scoping review of discrimination and health-related search terms revealed that there is a large number of publications on this topic (in total, 18,233 publications from 1966 to 2019, online search PubMed database), with a majority coming from the United States (43.6%), while only 5% were from Germany [55]. Full-text screening of 32 articles revealed that the concept of subjectively perceived discrimination and the survey instrument Everyday Discrimination Scale were most commonly used in studies [66]. This instrument captures daily-life experiences of disrespectful treatment, insult, or harassment in the first stage, and it then asks for possible reasons for being treated this way (e.g., ethnicity, gender, age, religion, or sexual orientation) in the second stage [55]. This instrument was further optimized within the scope of the cognitive pretesting, and a third part was added to capture subjectively perceived discrimination in health relevant settings; that is, in interaction with the health-sector (e.g., in hospitals) and with authorities (e.g., employment agency). The details on operationalization are presented in [53].

### 3.4. Public Health Reporting Concept and Development of Core Indicators

A total of 28 countries of the 45 countries who were invited took part in the survey on migration-related public health reporting. For the 17 non-responding countries, a standardized search strategy on their public health reporting was applied. Overall, 25 countries considered people with a migration background within their national public health reporting. Countries with a long history of migration, such as the United States, reported the health of migrant populations within their regular health reports. Focus reports were most often used (64%) as a reporting tool, followed by statistical online-databases (dashboards) (48%), and 44% of countries reported on migrant health as a cross-sectional topic. Most reports were released irregularly and did not focus on specific subpopulations among their migrant population. Some countries considered the specific needs of refugees and recent migrants (e.g., Sweden, Luxemburg, or Canada). The topics addressed within the reports were non-communicable diseases and mental health (90% each), health-related behaviour (76%), utilization of health-care services (71%), and infectious diseases (61%). Reporting was stratified by gender (85%), age (75%), and countries of birth (60%). Stratification for other migration-related characteristics was less frequent (duration of residence: 30%, residence status: 20%, migration generation: 20%, motives of migration: 20%, or socioeconomic position: 20%). These results show that comparability is lacking because countries defined *migration background* differently, and the quality of data diverges. 

To describe the health of people with migration background—adults (A), as well as children (C)—66 indicators were selected, of which 25 were defined as core indicators (Table 2). Further indicators and data sources to fill the indicators can be found in [61].

Besides the indicators, stratification characteristics were also developed within the process to account for the heterogeneity within the migrant population. Besides gender, age, and socioeconomic position, the indicators are recommended to be stratified by (a) migration background (non, second-generation, own history of migration), (b) country of birth, (c) duration of residence, (d) residence status, (e) migration motives, and (f) self-assessed German language proficiency [61].

## 4. Discussion and Recommendations

The IMIRA project was initiated to expand public health monitoring at the RKI to include people with a migration background and to further improve the public health reporting on migration and health. Within the project, we conducted two feasibility studies to identify strategies to include migrant populations into health interview and examination surveys. We also critically revised the existing concepts on capturing *migration background* and developed instruments to capture different dimensions of migration history, language skills, subjective feeling of belonging, social support, and discrimination to better describe the differences in health outcomes in the future. We additionally developed a set of core indicators for public health reporting on migrant health. From these findings and the lessons learned throughout the research process, we can derive the following components for a diversity-oriented public health monitoring. 

### 4.1. Use of Multilingual Materials

To bridge language gaps in health surveys, it is crucial to provide multilingual materials. In our feasibility study, the provision of bilingual information and the option to answer the questionnaire in translated versions were both well accepted; however, differences were observed between participants’ groups. In particular, participants with shorter duration of residence in Germany more often preferred the translated questionnaire [62]. Other studies have shown that multilingual materials can be effective when addressing language gaps; they also help to build trust and make the participants feel appreciated and welcome [62,67,68,69,70,71,72]. This also applies to the use of translated videos to explain examination procedures. This proved to be a good method to provide standardized information. This measure is planned to be used in future health examination surveys for all participants, independently of origin or spoken language. The results from our feasibility study “*Examination Survey*” showed that video-interpretation is a good tool for language mediation if the technical requirements are sufficiently met. This is in line with a review that found no differences in patient satisfaction between in-person and video-interpretation. The most important factor for patient satisfaction was for the interpreters to have been professionally trained [73].

### 4.2. Offering Different Modes of Survey Adminstration and Establishing Personal Contact

Most participants within the feasibility study “*Interview Survey*” participated online. Thus, it is important to offer modes of administration that are easily accessible for many participants. However, more participants with lower levels of education, as well as those reporting a moderate to very bad self-rated health, could be recruited during home visits. This result is also observable in other hard-to-survey population groups, such as older people [74]. This hints to diverse needs within the gross sample that need to be addressed to make the survey accessible for all participants. For feasibility reasons, it is useful to start with less costly modes of administration (e.g., online-surveys) and then proceed with written questionnaires to also include those who are willing to participate but unable to fill out an online questionnaire (e.g., if technical requirements are not fulfilled). Home visits including personal interviews are a gold standard for recruiting study participants, especially those that are hard-to-survey [24,72,75,76,77,78], but they are cost intensive and not always feasible. However, they are essential to obtain a comprehensive picture of the sample and of the participants’ health status and health needs because not offering home visits and personal interviews might lead to an exclusion of specific subgroups (e.g., those reporting a lower subjective health) [62,74]. During the German Health Interview and Examination Survey for Children and Adolescents (KiGGS) that took place from 2014–2017, home visits proved to be effective to motivate migrant families to participate and doubled the response [32]. The authors hypothesize that due to the personal contact, possible questions and doubts can be instantly addressed, and the details of the survey procedures can be explained. Other studies have shown that personal communication and building trust can increase response rates in migrant and other populations [17,41,71,72,79,80]. Our focus groups with face-to-face interviewers have shown that establishing personal contact was perceived as an effort in communication where the participants felt appreciated.

### 4.3. Considering Diversity Sensitivity

Another important lesson is that considering diversity sensitivity is a crucial component of including migrants into public health monitoring. This includes the development of materials that represent heterogeneity and appeal to the respective populations. In our case, focus groups to develop materials (e.g., invitation letters and study information) were helpful and resulted in well-accepted materials. In terms of questionnaire development, the utilization of cognitive pretesting proved to be an effective method to assure that the reception of concepts and categories was alike and comparable in the respective languages. Furthermore, it helped to achieve comparable answers regardless of the language. Based on the results of the pretesting, further optimization of the instruments was performed. We also observed that certain concepts (e.g., “community life”) gave rise to very different and partially contradicting associations. These differences arose from different background meanings of the term in different communities and could not be eliminated solely by an adapted translation. Consequently, this question was removed from the utilized questionnaire [53].

Diversity sensitivity is also crucial when considering data analysis and public health reporting on migrant health, which includes a more differentiated approach in considering migration status. Until recently, the use of *migration background* as a stratification variable was common to describe differences in health outcomes between Germans and “people with migration background” (including children of migrants) [81]. While *migration background* might be a helpful concept when only limited data on migration status are available, it does not allow a detailed description of factors associated with migrant health, also regarding possible interdependencies and correlations, such as the connection between residence status, duration of residence, or socioeconomic status [53,82,83]. Furthermore, the category *migration background* is often criticized because of its logic of attribution to others instead of self-disclosure [42]. This is accompanied by the problem of “othering”, the consolidation of the perception of people in this category as “strangers” or “others” even when they and/or their parents are German [42]. The current debate about the usability of this category in civil society, academic, and also official state institutions leads to a fundamental rethinking of its use and new recommendations regarding the differentiated assessment of migration status or the relevant factors in each context [43]. Thus, the concepts included in future health surveys should cover the health resources and challenges, as well as the health needs of all people under study and should thus reflect the diversity of the general population [53]. Training of staff is important to ensure diversity sensitivity during all steps of the research process [84], for example, by implementing diversity trainings focusing on the social justice perspective and aiming for antidiscrimination, inclusion, and participation of diverse staff [85,86]. In addition, taking time for critical reflection about diversity-sensitive reporting and communication creates awareness and ensures a non-discriminatory discourse on migrant health [87].

An important aspect of diversity-oriented research is the consideration of fundamental ethical principles throughout the research process and, above all, the do-no-harm principle. Thus, diversity sensitivity also means taking a responsible and discrimination-sensitive approach and having an awareness of possible risks and effects for researched population groups. In particular, the danger of misinterpretation due to overly broad categories, confounders, generalizations and the possible associated attributions, stigmatization, essentialization of (alleged) differences, and the emergence or consolidation of exclusion and discrimination should be considered in this context [84,88].

### 4.4. Participation of Migrants in Research

Another way of increasing diversity sensitivity in public health monitoring and improving acceptance of health surveys in migrant populations is the active inclusion of migrants in the research process. As described earlier, the conduct of focus groups helped to better understand effective strategies to reach and recruit migrants for surveys as well as to improve quality of materials. Enabling participation of the groups under study in the research process ultimately gives a better understanding of the groups’ needs and will not only result in higher response rates but also in a process of mutual learning. This implies for researchers a more comprehensive understanding of the living situation and conditions of the group under study, as well as a better understanding for research and its benefits for the population under study [17,18,72,89]. Participatory health research is especially valuable to initiate research in groups that are considered hard-to-survey or in marginalized populations and includes the participation of groups under study in all steps of the research process [72,79,80,90,91,92,93] with the goal to depict and eventually reduce health inequalities together [94]. However, participatory health research might not always be feasible in public health monitoring because it is resource intensive in terms of finances and time. However, Bach et al. described that even enabling participation in parts of the research process is an effective way to improve quality of data [89]. This could (for example) entail the establishment of a community advisory board, hiring diverse staff, and involving representatives of migrant groups in future concept evaluation and development. 

### 4.5. Establishing a Regular Migrant Public Health Monitoring System

The components that we have described are helpful tools to plan and conduct surveys and to adapt public health reporting on migrant groups. In 2021, we implemented most of them and started conducting a multimodal interview survey in six languages among migrant groups selected according to the group size in Germany, which will be an important survey to collect data on migrant health. Other components will be integrated in future interview and examination surveys to ensure the representation of (possibly) the whole population living in Germany. However, with our sampling frames, which are mostly telephone- or register-based, we do not include particularly vulnerable populations, such as migrants without a residence permit or migrants who work temporarily in Germany (e.g., labour migrants). Other sampling strategies are needed to capture the health situation and health-care needs of these groups, such as participatory research projects [79,89,90,91,92,93] or respondent-driven sampling [20,72,95,96,97]. A regular public health monitoring system needs to be established to continuously report on migrant health and their health-care needs, as well as to identify particularly vulnerable subgroups for intervention planning. Public health reporting on migrant health should use the format of focus reports to describe the health status of specific subpopulations (e.g., refugees or older migrants) but should include migration and health as a cross-sectional topic into regular public health reporting, such as by making use of online-databases (dashboards). A review from the European Region of the World Health Organization (WHO) has shown that only 23 of the 53 WHO member states had systematic approaches to collecting data on migrant health. However, many of them only focused on routine data that are collected as secondary data [98]. This reveals a substantial information gap that needs to be addressed and calls for a collaborative effort to establish European standards on migrant public health reporting that can be implemented nationally [99]. Implementing and sustainably funding an inclusive and diversity-oriented public health monitoring requires financial resources and political willingness and is crucial to address the growing diversity in our societies. 

## 5. Conclusions

Through the measures taken within the IMIRA project, we can derive the following components as fundamental to a diversity-oriented public health monitoring: (1) The use of multilingual and diversity-sensitive materials is crucial to bridge language gaps and build trust of participants. In addition, utilizing video-interpretation services to overcome language barriers and to obtain informed consent during examination surveys is an effective tool. (2) Offering different modes of interview administration (e.g., online, in writing, or face-to-face interviews) helps to include diverse groups of survey participants. Personal contact, such as during home visits, increases the likelihood of participation in general and in particular in hard-to-survey groups. (3) Increasing diversity sensitivity in materials, survey instruments, and in research staff can help to establish responsible and anti-discriminatory public health reporting on migration and health. (4) Increasing the participation of people with migration history in research and reporting (e.g., as staff or external advisors) helps to identify potential challenges and strategies for inclusion of migrant populations in public health monitoring and public health reporting. (5) A continuous effort and a regular migrant public health monitoring system to sustainably report on health (care) needs and resources are required. 

## Figures and Tables

**Figure 1 ijerph-19-00798-f001:**
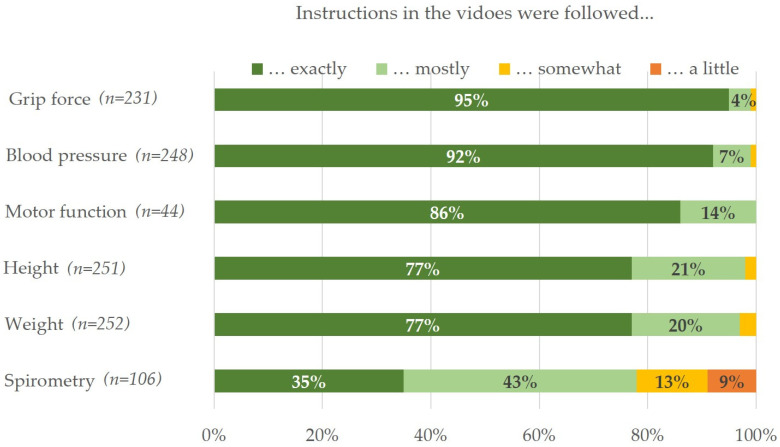
Evaluation of study personnel for how the participants followed instructions that were explained in the videos in terms of the different examinations.

**Table 1 ijerph-19-00798-t001:** Results on participation and sample composition in different administration modes of the IMIRA feasibility study “*Interview Survey*”, *n* = 1190 participants (adapted from [62]).

	Croatian		Polish		Romanian		Syrian		Turkish	
	*n*	%	*n*	%	*n*	%	*n*	%	*n*	%
Participants overall	178	100.0%	221	100.0%	109	100.0%	465	100.0%	217	100.0%
Response rate *	14.3%		19.9%		15.4%		24.3%		8.6%	
**Mode of administration**										
Online	149	83.7%	196	88.7%	76	69.7%	375	80.7%	155	71.4%
Telephone (hotline)	29	16.3%	25	11.3%	9	8.3%	22	4.7%	8	3.7%
Face-to-face	-	-	-	-	20	18.4%	41	8.8%	39	18.0%
Telephone (after obtaining number)	-	-	-	-	4	3.7%	27	5.8%	15	6.7%
**Language used**										
German	136	76.4%	108	48.9%	50	45.9%	89	19.1%	128	59.0%
Translation	42	23.6%	113	51.1%	59	54.1%	376	80.9%	89	41.0%
**Sample composition: online and telephone mode (hotline)**										
**Sex**										
Female	81	54.5%	118	53.4%	54	63.5%	176	44.3%	85	52.1%
Male	97	45.5%	103	46.6%	31	36.5%	221	55.7%	78	47.9%
**Age groups**										
18–44 years	88	49.5%	85	38.4%	47	55.3%	234	58.9%	88	54.0%
45–64 years	41	23.0%	89	40.3%	23	27.1%	119	30.0%	40	24.5%
65 years and older	49	27.5%	47	21.3%	15	17.6%	44	11.1%	35	21.5%
**Education** °										
Low	35	19.7%	24	11.0%	9	10.5%	108	27.3%	52	31.9%
Middle	80	44.9%	96	43.8%	33	38.8%	171	43.3%	68	41.7%
High	63	35.4%	99	45.2%	43	50.6%	116	29.4%	43	26.4%
**Self-rated health**										
Moderate/bad/very bad	44	24.7%	49	22.2%	16	18.8%	96	24.2%	53	32.5%
Good/very good	134	75.3%	172	77.8%	69	81.2%	301	75.8%	110	67.5%
**Sample composition: home visits (face-to-face, obtaining telephone number)**										
**Sex**										
Female	-	-	-	-	14	58.3%	31	45.6%	37	68.5%
Male	-	-	-	-	10	51.7%	37	54.4%	17	31.5%
**Age groups**										
18–44 years	-	-	-	-	14	58.3%	31	45.6%	13	24.1%
45–64 years	-	-	-	-	8	33.3%	25	36.8%	23	42.6%
65 years and older	-	-	-	-	2	8.4%	12	17.7%	18	33.3%
**Education** °										
Low	-	-	-	-	10	41.7%	20	29.4%	37	72.5%
Middle	-	-	-	-	11	45.8%	30	44.1%	11	21.6%
High	-	-	-	-	3	12.5%	18	26.5%	3	5.9%
**Self-rated health**										
Moderate/bad/very bad	-	-	-	-	7	29.2%	44	64.7%	32	59.3%
Good/very good	-	-	-	-	17	70.8%	24	35.3%	22	40.7%

* AAPOR Response Rate 1, calculated with the AAPOR Survey Outcome Rate Calculator 4 [63]. ° According to CASMIN (Comparative Analysis of Social Mobility in Industrial Nations) classification.

**Table 2 ijerph-19-00798-t002:** Core indicators developed within the IMIRA project to describe the health of people with migration background in public health reporting (adapted from [61]).

Topic	Indicator
**1. Promoting and strengthening health**	
**1.1 General health**	
Subjective health	Self-assessed general health (good to very good)
Chronic diseases (general)	12-month prevalence of chronic diseases in general
**1.2 Physical health**	
Cardiovascular disease	Lifetime prevalence of heart disease, including cardiac insufficiency and heart failure (self-reported medical diagnosis)
Heart disease (A)	
Stroke (A)	Lifetime prevalence of stroke (self-reported medical diagnosis)
Respiratory diseases	Lifetime prevalence of bronchial asthma (self-reported medical diagnosis)
Bronchial asthma	
Diabetes mellitus	Lifetime prevalence of diabetes mellitus (self-reported medical diagnosis)
**1.3 Mental health**	
Depressive disease (A)	Lifetime prevalence of a depressive illness (self-reported medical diagnosis)
Anxiety disorders	Lifetime prevalence of anxiety disorders (self-reported medical/psychotherapeutic diagnosis)
Psychological disorders (C)	Prevalence of psychological disorders in the last six months (total score from the Strengths and Difficulties Questionnaire)
**1.4 Infectious diseases**	
Tuberculosis	Tuberculosis cases among people born outside of Germany as a proportion of all tuberculosis cases
**2. Promoting health-conscious behaviour**	
**2.1 Nutrition-related behaviour and physical activity**	
Sporting (in)activity	Prevalence of sporting inactivity (no sports/very rarely)
Vegetable consumption	Daily vegetable consumption
Breastfeeding (C)	Proportion of children who have been exclusively breastfed for at least six months, as recommended by the World Health Organization
Body mass index (BMI)	Prevalence of overweight
	Prevalence of obesity
**2.2 Substance use/addiction**	
Tobacco use	Prevalence of current smoking (occasional to daily/regular)
Alcohol consumption	Prevalence of hazardous alcohol consumption (risk-related consumption)
**3. Promoting health-related resources and risk reduction**	
**3.1 Social and personal resources**	
Social support	A middle to high level of social support
**3.2 Migration-specific, psychosocial burdens**	
Experiences of discrimination	Experiences of discrimination (occasional to frequent)
**4. Promoting equal access to health-care services**	
**4.1 Utilization of preventive services**	
Vaccination (C)	Vaccination rates for the first and second measles vaccinations
Early detection examinations (C)	Full utilization of the U3 to U9 early detection examinations
Cervical cancer screening (A)	12-month prevalence of cervical cancer screening
Dental check-ups (C)	Adherence with the recommended utilization of dental check-ups
**4.2 Utilization of health-care services**	
Outpatient care (paediatrics, general) (C)	12-month prevalence of the utilization of outpatient paediatrics and general medical services
Outpatient care (general) (A)	12-months prevalence of the utilization of outpatient services from general practitioners

## Data Availability

The data presented in this study are available on request from the corresponding author. The data are not publicly available due to data protection restrictions.

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
