# Peer review of "Results and Strategies for a Diversity-Oriented Public Health Monitoring in Germany"

_ijerph, 2022, doi:10.3390/ijerph19020798_

Round 1

Reviewer 1 Report

This is an important descriptive study as it highlights the importance of cultural variations and perspectives in the design of research methods, particularly when the inquiry is centered on the experiences of people from various cultures. The soundness of the research design-sample selection, operationalization of key variables and their measurement, data collection techniques ,etc-is critical to the reliability and validity of the results. The authors offer some important suggestions and tools in promoting inclusivity and increased representation  of people from other cultures in research and data collection that promote sounder research designs and culturally responsive measurement and data collection.  The recommendations based on the findings offer opportunities for future researchers to be successful in cross-cultural research design and execution.

Overall, the manuscript can benefit from a thorough editing to be more   succinct.  The authors should consider either consider leaving out some of the more peripheral background information on the various studies or indicate them in a footnote. 

There are some few minor words and phrases that, if changed, would make the meaning clearer. For instance:

Line 47, suggest using the word, analysis. 

Line 113, suggest using the phrase "showing positive"

Line 145 and line 532, suggest using the phrase " hard to reach"

Line 158, suggest using the word, "conduct" instead of "realize

In the introduction section (starting on Line 30), the issue that Germany faces with becoming an increasingly multilingual and multicultural country is also true of other nations in Europe and other parts of the world. A statement that this issue that Germany faces is global would be helpful in this section, as it would engage or be of interest to researchers/readers beyond Germany

In the results table, the category EDUCATION is listed as "Low" , "Middle", "High".  Can the authors provide a specific measure of each? For example, is "Low"  referring to Less than High School Education? 

To highlight and emphasize the recommendations made in the Discussion section; It  is suggested that add "Recommendations" to the Discussion heading, "Discussion & Recommendations". 

In section, 4.5. Establishing a regular Migrant Public Health Monitoring System (starting on Line 497), the authors speak to some of the limitations of their study as the limitations posed by the available tools and resources. As part of their recommendations, could the authors elaborate on resources that other researchers should have in hand in order to successfully implement the suggestions/recommendations that are made by the authors.  What role should training play, if any, in preparing researchers to implement one or more of the recommendations made by the authors in order to enhance their cultural responsiveness and inclusivity in research?  What should the training program include, if the authors see a place for training? 

Author Response

Thank you for your valuable feedback. Please find our responses within the attachment.

Reviewer 2 Report

The paper fulfils the aims and scope of Plants. The article ‘Results and Strategies for a Diversity-Oriented Public Health Monitoring in Germany presented for review is interesting. The subject matter is of great social importance. However, some issues need to be clarified or supplemented. The comments are included below.

Title

The title is worded correctly and accurately reflects the content.

Abstract

The abstract is clear and adequate.

  1. Introduction

- The introduction serves multiple purposes. It introduces well to the subject of the article. Unfortunately, some fragments of the introduction are somewhat chaotic and it is not known whether they refer to literature research or to own research. Please verify this part of the work.

Minor another comments are provided below.

Line 70: Please check the correctness of the literature citation.

Line 83: Please check the correctness of the literature citation.

Line 96-97: Please check the correctness of the literature citation.

Line 104-105: Please check the correctness of the literature citation.

Line 107-108: Please check the correctness of the literature citation.

  1. Materials and Methods

Please indicate in the methodology how large was the research group? Was the same number of people involved in the first and second stages of the research?

Line 186: A comprehensive literature research… At this point, it is worth quoting the relevant literature.

  1. Results

The research results are correctly presented.

  1. Discussion

The discussion of the obtained results was carried out correctly.

  1. Conclusions

- The conclusions section is correct.

Others

- The literature is correct.

Author Response

(The authors gave the same response as above.)

Reviewer 3 Report

This is an interesting study looking at the importance of including diverse populations in public health data, particularly migrant populations.  My primary concern with the paper is that the authors do not situate the paper within a larger academic discourse.  As a result, I'm not sure where the paper contributes to the academic community.  Does this research advance scholarship on reaching migrant communities, or does the research contribute to creating more inclusive health data.  Without a literature review, I'm not sure what motivates the research and where the research contributes to the larger academic community and who is the audience for this paper.  '

I think the research methods are appropriate and yield useful results.  I also think the discussion of the methods is a bit confusing.  I was unsure how the videos fit into the research, until I reread the section several times.  If I understand correctly, the videos provide a method of self assessment for interviewees.  

Finally, I would suggest you proofread and maybe have a native English speaker edit your paper.  There are several grammar and style mistakes throughout the paper.  You employ passive voice several times throughout the paper, some of your word choices are awkward.  For example, you say Germany is an immigration country.  I would suggest you revise the statement to make it more readable.  Also, including headings in text (Background and results in the text of the abstract) is typically not a widely accepted approach to writing an abstract.  Some of your word choices are not just awkward, but are also incorrect.  For example, in line 48, "analyzes" should be "analysis."  A thorough proofread will help with these types of mistakes.  

I think the topic of migrant health is an extremely important research topic.  I found your approach very interesting.  I hope you will conduct a literature review to situate your research questions and conclusions into the academic literature and then edit the organization and style and grammar of the paper to help with readability.  I think once you address these concerns, you will have a very important contribution to the academic literature.  Best of Luck.  

Author Response

(The authors gave the same response as above.)

Round 2

Reviewer 3 Report

Thank you for addressing my comments.  I now understand where you plan on situating this article in the academic literature and how the videos fit into your study.  I think this is an interesting approach to studying the societal challenges faced by minority, immigrant communities.  I really enjoyed the article.